# Recycling of Retired Wind Turbine Blades into Modifiers for Composite-Modified Asphalt Pavements: Performance Evaluation

Peixin Li [1,2], Xiaodan Wang [2], Weijie Chen [2], Tao Yang [3], Xiaoya Bian [4,5] and Xiong Xu [4,5,*]

1    Guodian United Power Technology Co., Ltd., Beijing 100039, China; 20034652@chnenergy.com.cn
2    Guoneng United Power Technology (Baoding) Co., Ltd., Baoding 071000, China; 12101012@chnenergy.com.cn (X.W.); 12073608@chnenergy.com.cn (W.C.)
3    Wuhan Zhigu Advanced Technology Research Co., Ltd., Wuhan 430050, China; yt49@163.com
4    School of Civil Engineering and Architecture, Wuhan Institute of Technology, Wuhan 430073, China; wit_bianxy@hust.edu.cn
5    Hubei Provincial Engineering Research Center for Green Civil Engineering Materials and Structures, Wuhan Institute of Technology, Wuhan 430074, China
*    Correspondence: xxucea@wit.edu.cn

**Abstract:** With the rapid development of wind energy, large-scale disposal of retired wind turbine blades (rWTBs) has become a hotspot issue worldwide, especially in China. Currently, some practices have reused them in producing artworks, bus stations, concrete structures, etc., but their consumption and value are considered to be very low. Therefore, the recycling of rWTBs into asphalt pavement may be a good way to achieve the goals of large consumption and added value. On this basis, this study first obtained rWTBs crushed and ground into fine powders and then mechanically mixed with styrene–butadiene rubber after silane treatment for the final preparation of the powder modifier (R-Si-rWTB). Afterward, these modifiers were used to prepare composite-modified asphalt mixtures in combination with SBS. Through a series of structure and performance characterizations, the following valuable findings were reached: after the silane and rubber treatments, the microstructure of rWTBs became tougher and almost all of the fibers were coated by the rubber; the R-Si-rWTB modifier had a significant effect on improving the resistances of the asphalt mixture to moisture-induced damage, reaching 95.6%; compared to that of the virgin asphalt mixture (83.67%), the immersed residual Marshall stability of the 30R-Si-rWTB/70SBS asphalt mixture was higher, being between 86% and 90%; the rut depth development of 30R-Si-rWTB/70SBS was very close to that of 0R-Si-rWTB/100SBS, and their dynamic stabilities were close to each other, namely, 5887 pass/mm and 5972 pass/mm; and after aging, the resistances of the 30R-Si-rWTB/70SBS asphalt mixture to moisture and freeze–thaw damage improved. Overall, the value-added recycling of rWTBs into a modifier can contribute to better and more durable asphalt pavement.

**Keywords:** retired wind turbine blades; asphalt modifier; composite modification; asphalt mixture; performance evaluation

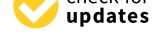



## 1. Introduction

With the rapid development of the wind energy industry in China, the quantity of retired wind turbine blades (rWTBs) will reach 8112 tons by 2025, and after 2025, the total number will increase rapidly and is expected to reach 412,784 tons by 2028 and about 715,664 tons by 2029 [1–3]. Since WTBs are mainly made of fiber-reinforced polymer (FRP) composites and most of their polymers are thermosetting materials [4–6], their main characteristics are non-biodegradability, high-temperature resistance, corrosion resistance, etc. [7–9]. Presently, simple treatment methods, such as stacking, landfill, and incineration, are still commonly adopted [10–12], but it is hoped that they will be reused

following the adoption of national policies on environmental protection and resource utilization [13]. Therefore, it is of great urgency and importance to consider how to dispose of rWTBs.

In order to fully realize the resource recycling and reutilization of rWTBs, some companies and researchers at home and abroad have applied their recycled products to different fields [14–17]. For example, the National Energy Group used rWTBs to make public transportation platforms; a Danish company made rWTBs into bicycle sheds by designing, cutting, and other means; a Dutch company turned them into play pools; a Chinese company powdered the blades for 3D printing, and the additive manufacturing was used to make them into some structures with low strength requirements; and a researcher from a university in China also tried to powder the blades for wall panel manufacture. Although these recycling methods show obvious social value, they still exhibit low levels of consumption and low added value, and it is difficult to achieve large-scale value-added applications through their adoption. For this reason, it is urgent and necessary to find other high-quality ways to reuse rWTBs.

To achieve the high-value and high-consumption reuse of rWTBs, they are here considered as potential materials that may enhance the engineering performance of asphalt pavements and address the associated issues in terms of storage and the environment. In recent years, there have been almost no studies related to the recycling and reuse of rWTBs in asphalt pavements; however, considering its material characteristics are close to those of FRP composites, some relevant studies in this aspect can be referenced to aid understanding. For example, Lin et al. [18] used GFRP as a filler to improve the engineering performance of an asphalt mixture and found that GFRP can improve the resistance of the asphalt mixture to rutting, fatigue, aging, peeling, etc., but is not amenable to low-temperature performance. Yang et al. [19] also found that FRP, as an asphalt reinforcement material, can not only improve the high-temperature performance of asphalt mixtures, including creep stiffness, rutting resistance, and creep-recovery behavior but can also enhance moisture-induced damage resistance. These similar studies demonstrated that FRP, as a filling material, can feasibly be applied in asphalt pavements for the enhancement of their performance characteristics. Therefore, it is believed that rWTBs can potentially be reutilized as an asphalt modifier to improve the properties of pavements.

As it is well known in the field of asphalt studies, SBS is a widely used polymer modifier that can contribute to improving the high- and low-temperature properties of asphalt binders and mixtures, but it is very expensive. Considering this, previously published studies have already investigated whether there are other cheaper materials that can be collectively adopted to replace the incorporation of SBS with no performance compromise in terms of the modified mixtures [20,21]. rWTBs normally consist of glass fiber and epoxy resin, which have been proven effective in improving the overall properties of asphalt pavements and are also considered to have the potential to reduce the use of SBS in pavement applications [22]. Therefore, the combined use of rWTBs and SBS may be necessary to advance the progress of high-quality asphalt pavements.

To achieve this goal, the current study innovatively proposes a new method for the value-added recycling of rWTBs into asphalt modifiers through mechanical crushing and grinding in association with further modification by silane and SBR. Further, this rWTB-based modifier will be examined both individually and in combined use with SBS to modify virgin asphalt in the preparation of modified asphalt binders and mixtures with different mixing proportions. The microscopic morphology and molecular structure of the modifiers will be analyzed, and the properties of the rWTB-modified mixtures will be checked through water immersion and freeze–thaw tests. Further, their engineering performances, especially moisture-induced damage resistance, will be evaluated after aging. Overall, value-added recycling of rWTB into modifiers can contribute to better serving more durable and cost-effective asphalt pavement.

## 2. Materials and Methods

### 2.1. Raw Materials

2.1.1. Virgin Bitumen

The asphalt binder used in this study was virgin bitumen (Pen. 70 grade), which was supplied from a local factory. The main physical properties were measured according to the results of standard tests, which are presented in Table 1.

**Table 1.** Test results of main physical properties of asphalt binder.

| Parameter | Test Result | Requirement | Standard |
|---|---|---|---|
| Penetration (0.1 mm) | 65 | 60–80 | ASTM D5 |
| Softening point (°C) | 48.2 | ≥43 | ASTM D36 |
| Viscosity at 135 °C (Pa·s) | 0.46 | ≤3 | ASTM D4402 |
| Ductility at 15 °C (cm) | >100 | >100 | ASTM D113 |

2.1.2. Recycled Wind Turbine Blades (rWTBs)

The rWTBs were the end-of-life composite materials from a wind farm in China, which were mainly composed of glass fiber and epoxy resin. The physical appearance of the rWTB is shown in Figure 1. Prior to use, the blade was crushed into powder in different sizes; powder sized less than 0.3 mm was selected for preparing modifiers.

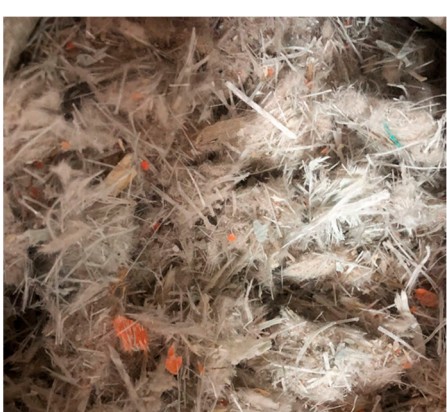

**Figure 1.** Physical appearance of rWTBs.

2.1.3. Silane Coupling Agent

The silane coupling agent was an analytically pure γ-aminopropyl triethoxysilane marked with KH550, with a molecular formula of $C_9H_{23}NO_3Si$ and a structural formula of $H_2NCH_2CH_2CH_2Si(OC_2H_5)_3$. Its main physical properties are listed in Table 2.

**Table 2.** Main physical properties of KH550.

| Property | Description |
|---|---|
| Purity | Analytically pure |
| Appearance | Colorless and transparent liquid |
| Gravity (25 °C, g/cm$^3$) | 0.946 |
| Boiling point/°C | 217 |
| pH | >7 (after hydrolysis) |
| Solubility | Soluble in water |

2.1.4. Styrene–Butadiene Rubber

Polymerized styrene–butadiene rubber (SBR) is composed of irregular copolymerized butadiene and styrene. Its physical and processing properties are close to those of natural rubber, and its wear, heat, aging, and water corrosion resistances are even better than

those of natural rubber. In addition, with its high flexibility, SBR absorbs the oil in asphalt and develops into a mesh structure, which can effectively improve the high- and low-temperature stress dissipation capacities of asphalt. In this study, SBR was used to prepare an rWTB-based asphalt modifier with the physical characteristics provided in Table 3.

**Table 3.** Main performance indicators of the used SBR.

| Item | Unit | Description |
|------|------|-------------|
| Appearance | \ | White particle |
| Moisture content | % | ≤0.5 |
| Relative molecular weight | $10^4$ | 20~30 |
| Styrene content | % | 26~35 |
| Rubber content | % | ≥90 |

### 2.1.5. SBS

In this study, SBS was used for the composite modification of asphalt bitumen with the prepared rWTB-based modifier. The used SBS had a linear structure and was purchased locally; its main controlled indicators are shown in Table 4.

**Table 4.** Main controlled indicators of the used SBS.

| Item | Unit | Description |
|------|------|-------------|
| Appearance | \ | White particle |
| S/B proportion | \ | 40/60 |
| Oil-filling content | % | 0 |
| Volatile content | % | ≤0.7 |
| Ash content | % | ≤0.20 |
| Elongation at break | % | ≥730 |
| Melt flow rate | g/10 min | 0.10~5.00 |

### 2.1.6. Sulfur

Sulfur was supplied from a local factory, with some of its main technical indices presented in Table 5. In this study, sulfur was used in situ to enhance the elasticity of the SBR in the asphalt binder by premixing it into rWTB-based blends to prepare a high-performance modifier.

**Table 5.** Main technical indices of sulfur.

| Item | Unit | Description |
|------|------|-------------|
| Appearance | \ | Yellow powder |
| Density | $g/cm^3$, at 23 °C | 1.98 |
| Boiling point | °C | 460 |
| Burning point | °C | 270 |
| Melting point | °C | 115 |
| Ash content | % | 0.003 |
| Solubility | \ | Water-insoluble |

### 2.1.7. Aggregates

The natural aggregates and mineral fillers were limestone, which was used to prepare the asphalt mixtures in this study. The primary properties of the coarse and fine aggregates are shown in Table 6.

**Table 6.** Technical values of the parameters of the used aggregates.

| Category | Test Item | Test Result | Requirement |
|---|---|---|---|
| Coarse (>2.36 mm) | Apparent density, g/cm$^3$ | 2.864 | $\geq$2.60 |
| | Absorption (%) | 1.71 | $\leq$2.0 |
| | Crushing value (%) | 16.2 | $\leq$26 |
| | Robustness (%) | 3.2 | $\leq$12 |
| Fine (<2.36 mm) | Apparent density, g/cm$^3$ | 2.749 | $\geq$2.50 |
| | Robustness (%) | 2.6 | $\leq$12 |
| | Sand equivalent (%) | 83.3 | $\geq$60 |

*2.2. Preparation of rWTB-Based Asphalt Modifiers*

To optimally prepare rWTB-based asphalt modifiers, four different kinds of rWTB modifiers were designed and prepared using physical and chemical methods for performance comparisons in this study.

### 2.2.1. rWTB Modifier

The first asphalt modifier was prepared via a physical method of mechanical crushing and grinding processing. The detailed procedures were as follows: (1) The rWTB was mechanically crushed into flakes of approximately 5–13 mm using a crusher. (2) These crushed rWTB flakes were ground into powders using a grinder for at least 5 min. (3) The obtained powders were passed through a screen to obtain particles sized below 0.3 mm for use as the asphalt modifier. This modifier was labeled the rWTB modifier.

### 2.2.2. Si-rWTB Modifier

The second asphalt modifier was obtained via a silane surface treatment method as follows: (1) Anhydrous ethanol and deionized water were mixed in a 9:1 mass ratio to produce a mixed solvent. (2) KH550 was added to the mixed solvent in a mass ratio of KH550:ethanol = 1:9, and the mixes were slowly stirred with a glass rod to prepare the hydrolysate. (3) A certain amount of rWTB powder was added to the prepared hydrolysate, which was mechanically stirred for 30 min at 80 °C. (4) The blends were cured in an oven at 105 °C for 2 h; after grinding, the modified powders were used for bitumen modification. The powder was labeled Si-rWTB.

### 2.2.3. R-rWTB Modifier

The third asphalt modifier was prepared by directly mixing SBR with rWTB in a mechanical mixer. The preparation steps were as follows: (1) Certain amounts of rWTB and SBR, in a mass ratio of 1:0.3, were manually mixed. (2) The blends were added into a chamber for mechanical mixing at 60 °C for 5 min at a shearing speed of 50 rpm. (3) The mixes were then collected for crushing and grinding to obtain particles smaller than 0.3 mm after screening. These powders, labeled R-rWTB, were used for bitumen modification.

### 2.2.4. R-Si-rWTB Modifier

The fourth asphalt modifier was also prepared by mixing SBR with Si-rWTB in a mechanical mixer. This preparation process was similar to that introduced in Section 2.2.3. The difference was that the base powder that was modified was the Si-rWTB prepared from the process in Section 2.2.2. This modifier was labeled R-Si-rWTB.

*2.3. Preparation of Different rWTB-Modified Asphalts*

First, virgin bitumen was heated to 165 °C. Then, 5% rWTB, by weight of the virgin binder, was mixed with a molten binder and sheared at 165 °C for 30 min at a rate of 2000 rpm. Subsequently, the rWTB-modified asphalt binder was sheared with an increased rate of 4000 rpm and blended for a further 30 min. Their modified asphalt binders were rWTB, Si-rWTB, R-rWTB, and R-Si-rWTB.

### 2.4. Preparation of Different rWTB/SBS-Modified Asphalt

First, virgin bitumen was heated to 165 °C. Then, 5% SBS, 3.5% SBS + 1.5% rWTB, 1.5% SBS + 3.5% rWTB, and 5.0% rWTB, by weight of the virgin bitumen, were blended into the virgin bitumen and mixed at a rate of 500 rpm for 30 min. Subsequently, the blends were sheared at 170 °C for 40 min at a rate of 4000 rpm to prepare different rWTB/SBS-modified asphalt binders.

### 2.5. Preparation of rWTB/SBS-Composite-Modified Asphalt Mixtures before and after Aging

For most highway construction in China, AC-20-type aggregate gradation (Figure 2) is widely used in the paving of the middle layer. According to the Marshall mix design, the optimal asphalt content (OAC) was determined to be 4.3% through standardized tests. Following this, different rWTB/SBS-composite-modified asphalt mixtures were prepared at 170 °C with a mixing duration of 100 s.

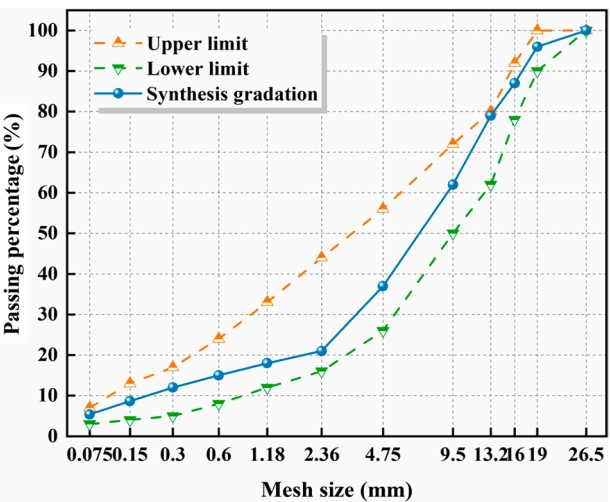

**Figure 2.** AC-20-type aggregate gradation.

After the modified asphalt mixtures were obtained, short- and long-term aging processes were conducted. For short-term aging, the mixtures were placed in an oven at 135 °C for 4 h under forced ventilation. For long-term aging, the mixtures were first molded into specimens after short-term aging and then were cooled at room temperature for more than 16 h. Once demolded, they were placed in an 85 °C oven for 5 days under forced ventilation.

### 2.6. Test Methods

#### 2.6.1. Scanning Electron Microscopy (SEM)

To understand the differences in the microstructures of the rWTB, Si-rWTB, and R-Si-rWTB modifiers, SEM images were captured after the treatments. The following steps were conducted: (a) The samples were sprayed with gold at a high vacuum pressure. (b) The samples were placed and fixed into a test chamber. (c) Images were captured at 500× and 5k× under different sets of test parameters. From the microstructural results, the changes in the surface of rWTB particles were observed to verify whether silane could organically modify the rWTB surface and whether the rubber could be better coated onto the Si-rWTB surface.

#### 2.6.2. Fourier Transform Infrared Spectroscopy (FTIR)

We employed a Nicolet 6700 FTIR spectrometer to verify the changes in the molecular structure of the prepared modifiers, including whether the silane was grafted onto the rWTB surface and whether the SBR coating was removed from the Si-rWTB surface via mechanical processing. Prior to the test, small modifier particles were mixed with ground KBr powders to prepare sheet specimens after pressing. During the test, the sheet specimen

was placed to the sample position in the spectrometer chamber and then tested under the following the conditions: resolution of 4 cm$^{-1}$, wavenumber range of 4000–400 cm$^{-1}$, and scanning time of 16.

### 2.6.3. Immersed Marshall Test

The immersed Marshall test was used in this study to evaluate the effect of the different rWTB modifiers on the moisture-induced damage resistance of the asphalt mixture. Except for the unaged mixtures, the aged mixtures were tested to indicate if the aging negatively impacted the moisture-induced damage of the modified asphalt mixtures. In accordance with JTG E20-2011 [23], the unaged and aged Marshall specimens were immersed in a 60 °C water bath for 30 min and 48 h, and then the residual Marshall stability values were collected after loading at 50 mm/min. The calculation formula is presented in Equation (1):

$$MS_r = \frac{MS_a}{MS_b} \times 100\% \tag{1}$$

where $MS_r$ is the residual Marshall stability after immersion, %; $MS_a$ is the Marshall load after immersion at 60 °C for 48 h, kN; and $MS_b$ is the Marshall load after immersion at 60 °C for 30 min, kN.

### 2.6.4. Freeze–Thaw Splitting Test

Similar to the immersed Marshall test, the freeze—thaw splitting test was adopted in this study to comparatively evaluate the moisture-induced damage of the asphalt mixtures with the incorporation of rWTB modifiers before and after aging. In accordance with JTG E20-2011, the Marshall samples were prepared by compacting each side 50 times, and then approximately 10 mL of water was added to each sample in plastic bags that were closed. Before the test, these samples were preconditioned in a refrigerator at −18 °C for 16 h and then placed in a water bath at 60 °C for 24 h. After this, the samples were tested at a loading rate of 50 mm/min to record the maximum load for the calculation of the splitting strength. On this basis, the freeze–thaw splitting strength ratio (TSR) was calculated to characterize the residual resistance of the samples to moisture-induced damage following Equation (2):

$$\text{TSR} = \frac{ITS_1}{ITS_0} \times 100\% \tag{2}$$

where TSR (%) is the freeze–thaw splitting strength ratio of the asphalt mixture; $ITS_0$ is the splitting strength of the asphalt mixture without undergoing freeze–thaw; and $ITS_1$ is the splitting tensile strength of the asphalt mixture after one freeze–thaw cycle.

### 2.6.5. Wheel Tracking Test (WTT)

The WTT is used to examine the resistance of asphalt mixtures to high-temperature deformation. According to JTG E20-2011, 300 mm × 300 mm × 50 mm rut specimens were prepared through a mechanical rolling method, placed into the test chamber of the rut device, and kept at 5 h at 60 °C. During the test, the specimen was rolled back and forth at a speed of 42 passes/min under a wheel load of 0.7 MPa for 60 min. After completion, the rut depth data were collected to calculate the dynamic stability (DS) as per Equation (3). With an increase in DS, the resistance of the asphalt mixtures to the high-temperature deformation was enhanced.

$$\text{DS} = \frac{15N}{d_2 - d_1} \tag{3}$$

where DS is the dynamic stability of the asphalt mixture, pass/mm; $d_1$ and $d_2$ are the rut depths of the asphalt mixture at 45 and 60 min, respectively, mm; and $N$ is the back-and-forth rolling speed of the test wheel, usually 42 passes/min.

## 3. Results and Discussion

### 3.1. Microstructure of rWTB-Based Asphalt Modifiers

Figure 3 displays the microstructures of different rWTB-based asphalt modifiers. As shown in Figure 3a, the overall rWTB microstructure presented fragments in fibrous structures, where smaller fragments were randomly piled on the surface. Figure 3b shows that after modification with the silane coupling agent, the Si-rWTB surface was relatively rough, and the interfaces were relatively blurred, indicating that the silane coupling agent was successfully coated on the rWTB surface. Furthermore, Figure 3c shows that as the SBR rubber was mixed, the R-Si-rWTB surface was tougher, and almost all fibers were embedded in the rubber with no interfaces, which demonstrated that the rubber firmly remained on the Si-rWTB surface, even after mechanical processing.

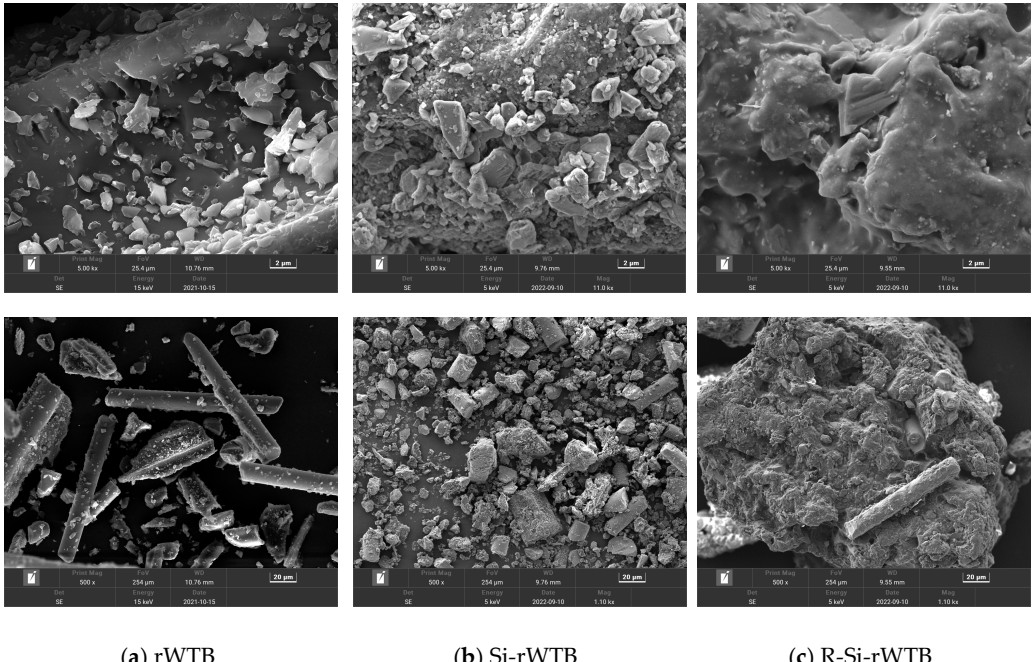

(**a**) rWTB            (**b**) Si-rWTB            (**c**) R-Si-rWTB

**Figure 3.** The microstructures of different rWTB-based asphalt modifiers.

From the above analysis, we summarize the principle of the microstructural changes that occurred during the preparation process of the rWTB-based asphalt modifier in Figure 4. During the preparation process, the microstructure of the rWTB modifier mainly evolved in the following three aspects: (a) the exposed smooth surface of the glass-fiber-containing resin was rich in Si-OH; (b) the surface was modified through the hydrolysis product of the KH550 end-amino silane coupling agent, with partial surface changes owing to Si-OH structures to the terminal amino silane structures; and (c) through thermal-mechanical mixing, SBR was thoroughly coated onto the Si-rWTB surface with the formation of new core–shell structure with almost no weak interfaces.

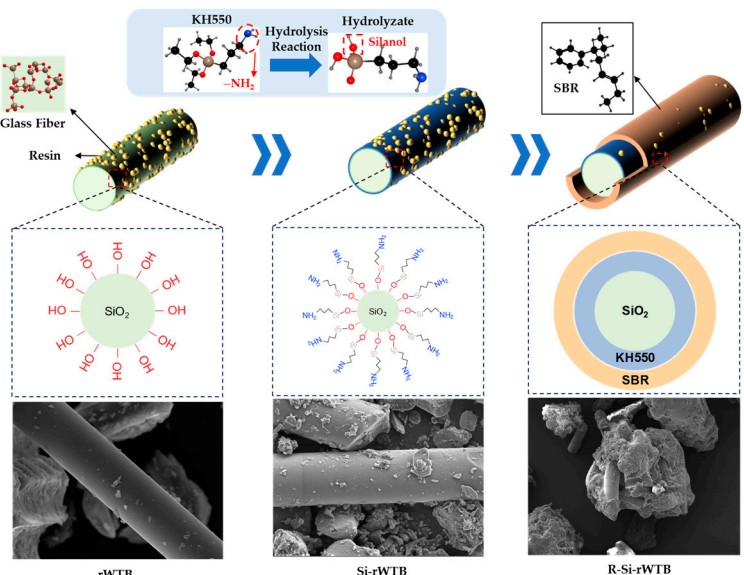

**Figure 4.** Schematic diagram of the microstructural changes that occur during the process of preparing rWTB-based asphalt modifiers.

### 3.2. Molecular Structure of Different rWTB Asphalt Modifiers

Figure 5 shows the FTIR results of the rWTB modifier before and after surface treatments. Additionally, the corresponding attributions of the main characteristic peaks are presented in Table 7. Curves a, b, and c all have some identical characteristic absorption peaks, such as the vibration absorption of the Si-OH peak at 3623 cm$^{-1}$, the asymmetric vibration absorptions of the Si-O-Si peak at 1132 cm$^{-1}$ and 1022 cm$^{-1}$, the symmetric vibration absorption of the Si-O-Si peak at 877 cm$^{-1}$, the bending vibration absorption of the Si-O-Si peak at 701 cm$^{-1}$ and 458 cm$^{-1}$, the vibration absorption of N-H from the -NH$_2$ peak at 3527 cm$^{-1}$ and 3456 cm$^{-1}$, and the asymmetric and symmetrical vibration absorptions of the methylene -CH$_2$- peak at 2938 cm$^{-1}$ and 2851 cm$^{-1}$. Compared with curve a, the vibration absorption peak intensities of silicon hydroxyl in curve b, the amino group, and methylene are strongly enhanced, but the methyl -CH$_3$ in the KH550 molecule does not appear, indicating that the hydrolysis reaction of KH550 led to changes in its structure. In addition, in curve b, the Si-O-Si vibration peaks at 1132, 1022, and 877 cm$^{-1}$ are more intense than those in curve a. This indicated that the silicon hydroxyl group on the rWTB surface reacted with the hydrolysis of KH550 to form a hydrogen bond, and the dehydration condensation reaction occurred during the curing process to form a Si-O-Si covalent bond, thus enhancing the characteristic peak of the Si-O-Si chemical bond. Overall, this further showed that KH550 was successfully grafted onto the rWTB surface. Moreover, curve c has an SBR characteristic absorption peak at 967 cm$^{-1}$, indicating that SBR was successfully coated onto the Si-rWTB surface.

**Table 7.** The attributions of the main characteristic peaks in Figure 5.

| Attribution | Wavenumber (cm$^{-1}$) | Vibration Type |
|---|---|---|
| Si-OH | 3623 | Stretching |
| -NH$_2$ | 3527, 3456 | Stretching |
| -CH$_2$- | 2938 | Asymmetric stretching |
| -CH$_2$- | 2851 | Symmetric stretching |
| Si-O-Si | 1132, 1022 | Asymmetric stretching |
| -CH = CH- | 967 | Out-of-plane bending |
| Si-O-Si | 877 | Symmetric stretching |
| Si-O-Si | 701, 458 | Bending |

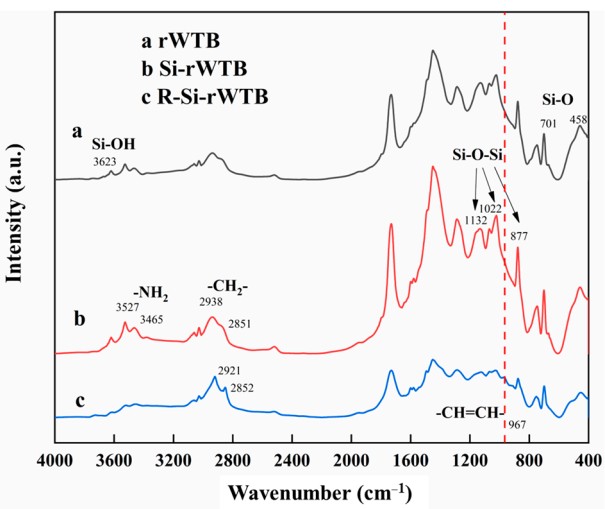

**Figure 5.** FTIR results of the rWTB modifier before and after surface treatments.

### 3.3. Effect of rWTB-Based Asphalt Modifiers on Marshall Stability before and after Immersion

The effects of the different rWTB modifiers on the Marshall stability of the asphalt mixture before and after immersion are presented in Table 8 and Figure 6. Compared with the untreated rWTB, the MS and $MS_1$ of the asphalt mixtures containing rWTB modified via the different surface treatments were both higher, indicating that the surface treatments with rWTB increased the interface strength between the asphalt binders and aggregates for the increase in the water stability of the asphalt mixture. Additionally, compared with that of the virgin asphalt mixture (83.67%), the residual stabilities of the asphalt mixtures containing rWTB, Si-rWTB, R-rWTB, and R-Si-rWTB were 87.28%, 90.37%, 93.49%, and 95.58%, respectively, indicating that the R-Si-rWTB modifier, as the target, considerably increased the water stability of the asphalt mixture after immersion.

**Table 8.** Results of the immersed Marshall stability of different rWTB-modified asphalt mixtures.

| Mixture | MS/kN | $MS_1$/kN | MS Increase/% | $MS_1$ Increase/% |
|---|---|---|---|---|
| | 30~40 min | 48 h | 30~40 min | 48 h |
| VB | 9.31 | 7.79 | 0 | 0 |
| rWTB | 11.14 | 9.72 | 20 | 25 |
| Si-rWTB | 11.22 | 10.14 | 21 | 30 |
| R-rWTB | 11.36 | 10.62 | 22 | 36 |
| R-Si-rWTB | 11.72 | 11.20 | 26 | 44 |

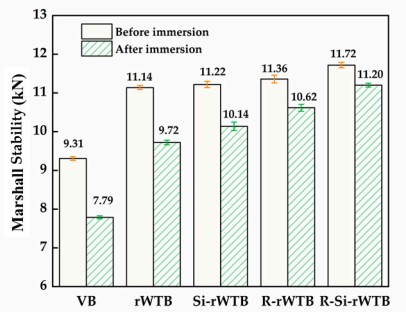
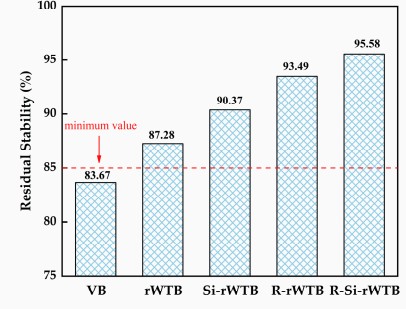

**Figure 6.** Effects of different rWTB modifiers on water stability of the asphalt mixture.

### 3.4. Moisture-Induced Damage Results of R-Si-rWTB/SBS-Composite-Modified Asphalt Mixtures

3.4.1. Immersed Marshall Test Result

Figure 7 displays the Marshall stability and residual stability of the R-Si-rWTB/SBS-composite-modified asphalt mixtures before and after immersion. The individual use of the SBS or R-Si-rWTB modifier and the hybrid use of the R-Si-rWTB and SBS modifiers effectively increased the Marshall stability of the virgin asphalt mixture before and after immersion. Compared with that of the virgin asphalt mixture (83.67%), the residual Marshall stability values of the modified asphalt mixtures after immersion were relatively higher, being roughly between 86% and 96%, indicating that the sole use of the R-Si-rWTB modifier or the hybrid use of R-Si-rWTB and SBS can considerably increase the immersed stability of a virgin asphalt mixture.

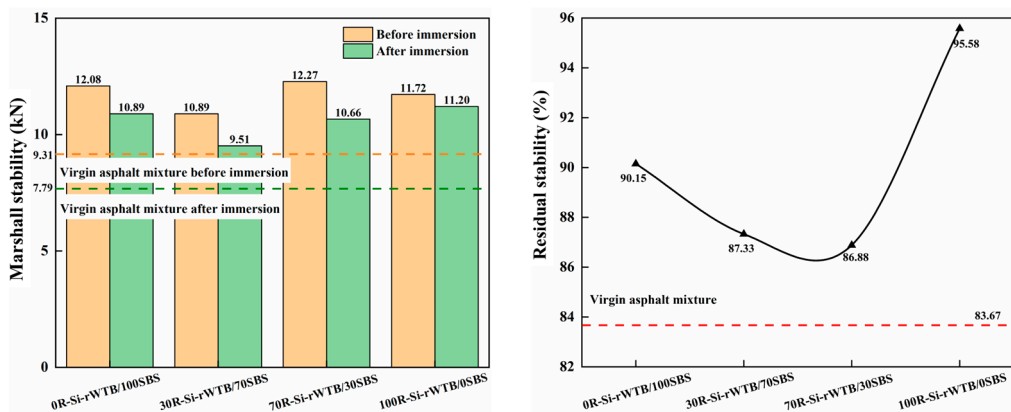

**Figure 7.** Marshall and residual stabilities of R-Si-rWTB/SBS-composite-modified asphalt mixtures before and after immersion.

3.4.2. Freeze–Thaw Splitting Test Result

Figure 8 displays the freeze–thaw splitting strength and the strength ratio of the R-Si-rWTB/SBS-composite-modified asphalt mixtures. The use of SBS or R-Si-rWTB alone and the combination of R-Si-rWTB and SBS effectively increased the splitting strength of the virgin asphalt mixture before and after freeze–thaw. Compared with that of the virgin asphalt mixture (0.81 MPa), the freeze–thaw splitting strength ratio of the modified asphalt mixtures were relatively higher, roughly between 93% and 96%, indicating that the combination of R-Si-rWTB or the sole use of R-Si-rWTB or SBS considerably increased the resistance of the virgin asphalt mixtures to the influence of freezing and thawing.

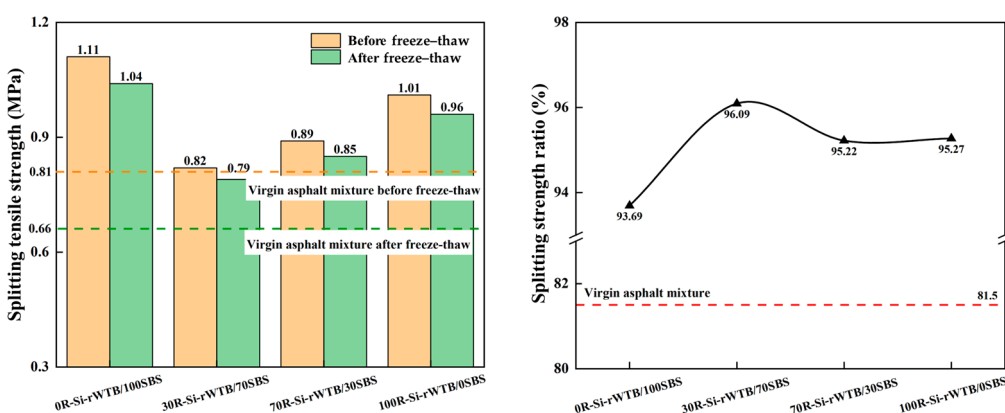

**Figure 8.** Freeze–thaw splitting strength and strength ratio of R-Si-rWTB/SBS-composite-modified asphalt mixtures.

### 3.5. High-Temperature Performance Results of R-Si-rWTB/SBS-Composite-Modified Asphalt Mixtures

Figure 9 shows the rut depth and dynamic stability of the different R-Si-rWTB/SBS-composite-modified asphalt mixtures. With an increasing R-Si-rWTB proportion, the rut depth of the modified asphalt mixture increased to some degree. Correspondingly, the rut depth at 60 min increased from approximately 0.5 mm to approximately 2.0 mm, and the dynamic stability decreased from 5972 to 3559 passes/mm, which is higher than that of the virgin asphalt mixture (1890 passes/mm). In addition, the rut depth development trend in the 30R-Si-rWTB/70SBS mixture was close to that of the 0R-Si-rWTB/100SBS mixture; correspondingly, their dynamic stabilities were close to each other. These results demonstrated that the replacement of SBS with 30% R-Si-rWTB did not notably affect the high-temperature stability of the original SBS-modified asphalt mixture.

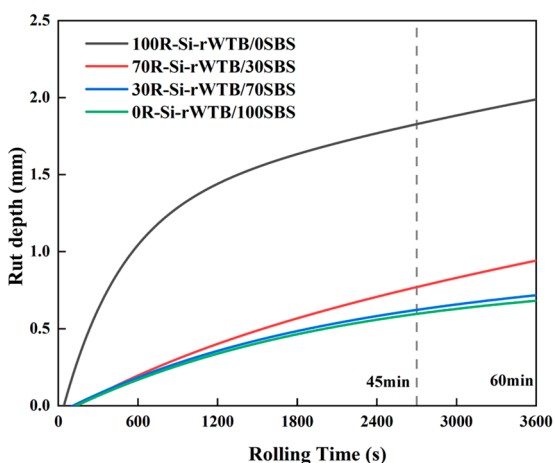 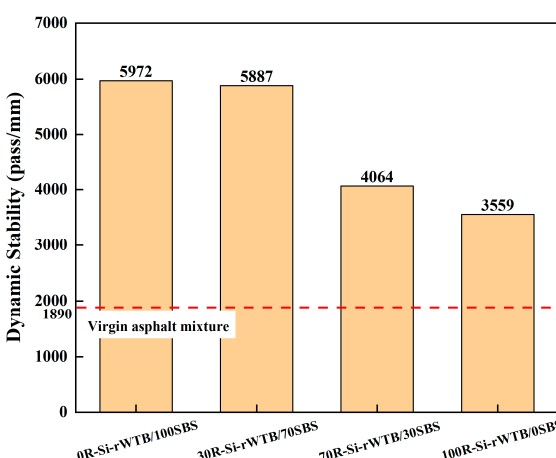

**Figure 9.** Rut depth and dynamic stability of different R-Si-rWTB/SBS -composite-modified asphalt mixtures.

### 3.6. Effect of Aging on Moisture-Induced Damage of R-Si-rWTB/SBS-Composite-Modified Asphalt Mixtures

#### 3.6.1. Immersed Marshall Test Result

Figure 10 reveals the influence of aging on the Marshall stability of different R-Si-rWTB/SBS-composite-modified asphalt mixtures. The Marshall stability of the modified asphalt mixtures increased to some degree after aging, especially that of the R-Si-rWTB/SBS-composite-modified asphalt mixture, which had a residual stability of 140–150%. This occurred because thermo-oxidative aging mainly leads to the volatilization and reaction condensation of the lower-molecular-weight substances in an asphalt binder, which increases the adhesion of the modified binder to aggregates. In addition, R-Si-rWTB was composed of epoxy-coated glass fiber as its core and sulfur-mixed rubber as its shell, which can produce the capillary adsorption effect of raw glass fiber on asphalt molecules and increase the flexibility and elasticity of the cross-linked rubber and non-crosslinked rubber in asphalt. After aging for 5 days, the glass fiber part of R-Si-rWTB continuously promoted the adsorption of asphalt molecules to the surface of the aggregates, whereas the rubber part maintained high flexibility and elasticity at the bonding interface between the aggregates.

Overall, the 5-day aging positively affected the engineering performance of the asphalt mixture. As aging time increases, the light components of the asphalt molecules no longer sufficiently migrate to the aggregate surface. The asphalt binder at the bonding interfaces hardens after aging, where the binder's dimensions gradually reduce to cause stress shrinkage, further resulting in the continuous deterioration in the engineering performance of modified asphalt mixtures.

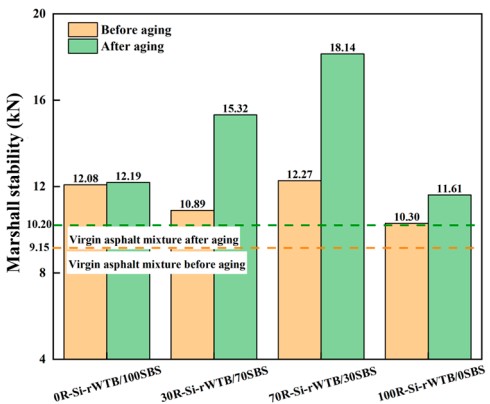
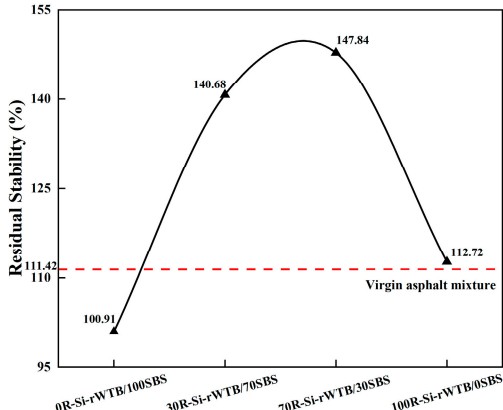

**Figure 10.** Effect of aging on Marshall stability of different R-Si-rWTB/SBS-composite-modified asphalt mixtures.

### 3.6.2. Freeze–Thaw Splitting Test Result

Figure 11 shows the effect of aging on the freeze–thaw splitting strength and the residual strength ratio of different R-Si-rWTB/SBS-composite-modified asphalt mixtures. After aging, the freeze–thaw splitting strength of the virgin asphalt mixture reduced from 0.81 MPa to 0.64 MPa, and the corresponding residual strength ratio was 79.83%; however, the freeze–thaw splitting strength of the different R-Si-rWTB/SBS-composite-modified asphalt mixtures tended to increase, particularly that of the 30R-Si-rWTB/70SBS-composite-modified asphalt mixture. The residual strength ratio of the 30R-Si-rWTB/70SBS-composite-modified asphalt mixture was 169.06%. This indicated that the use of 30R-Si-rWTB modifier alone or in combination with SBS modifiers can effectively increase the residual stability of aged asphalt mixtures after freeze–thaw cycles.

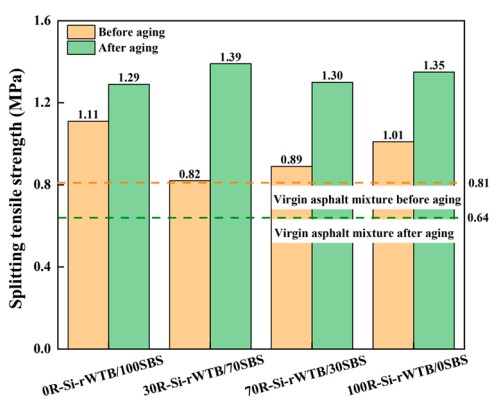
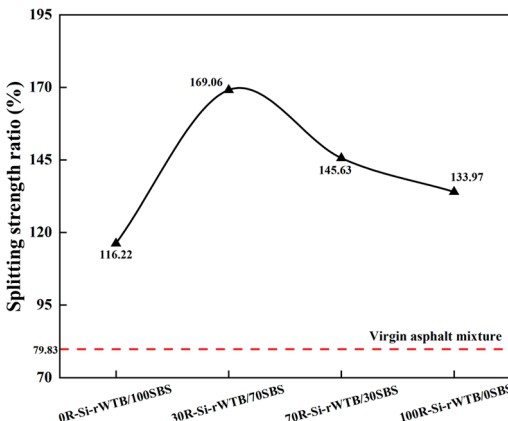

**Figure 11.** Effect of aging on the freeze–thaw splitting strength and residual strength ratio of different R-Si-rWTB/SBS-composite-modified asphalt mixtures.

### 4. Conclusions

This paper proposed a new method for converting rWTB to a high-value asphalt modifier through mechanical crushing and grinding, in association with further modifications with silane and SBR. In addition to the sole use of this rWTB-based modifier, it was considered in combination with SBS to jointly modify virgin binder for the preparation of composite modified asphalt binders and mixtures in different mixing proportions. With aging, the engineering performance—especially moisture-induced damage resistance—of the resulting asphalt mixtures was evaluated. Our main conclusions are as follows:

- The SEM and FTIR results indicated that, after silane and rubber treatments, the silane was successfully grafted onto the rWTB surface to increase compatibility with organic

substances; additionally, the microstructure in the presence of rWTB was tougher, and almost all fibers were inside the rubber with no interfaces.

- The moisture stability results demonstrated that the R-Si-rWTB modifier contributed to substantially increasing the resistance of the asphalt mixture to moisture-induced damage, increasing the residual Marshall stability from 83.67% to 95.6% after water immersion. When mixed with 70% SBS, the residual Marshall stability remained at 87.33%.
- The freeze–thaw test results showed that the splitting strength of the virgin asphalt mixture was higher both before and after freeze–thaw treatment when R-Si-rWTB was added; furthermore, the freeze–thaw splitting strength ratio of the modified asphalt mixtures remained higher, roughly between 93% and 96%, in comparison with that (81.5%) of the virgin mixture, showing that the R-Si-rWTB modifier effectively increased the resistance of the asphalt mixtures to freeze–thaw damage.
- The rut test results showed that the rut depth development of the 30R-Si-rWTB/70SBS asphalt mixture was close to that of the 0R-Si-rWTB/100SBS asphalt mixture, and their dynamic stabilities were also close to each other (5887 and 5972 passes/mm, respectively), indicating that 30% R-Si-rWTB did not notably affect the resistance of the asphalt mixture to high-temperature deformation.
- The moisture and freeze–thaw test results suggested that the resistances of the 30R-Si-rWTB/70SBS asphalt mixture to moisture and freeze–thaw treatment increased after a short aging duration.

Overall, this study provides a novel recycling and reuse method to convert rWTB into a value-added modifier that enhances the overall engineering performance of asphalt pavement. The developed modifier effectively replaced some of the SBS in the asphalt binder by increasing the durability of the mixtures to moisture-induced and aging damage while reducing cost savings and protecting the environment. The results indicated that the prepared modifier increased the resistance of asphalt pavement to moisture and freeze–thaw environments, demonstrating its suitability for different applications.

**Author Contributions:** Conceptualization, P.L. and X.X.; data curation, P.L., X.W. and T.Y.; formal analysis, P.L., W.C. and X.X.; funding acquisition, P.L.; investigation, P.L., X.W. and X.X.; methodology, P.L., X.W. and X.B.; resources, P.L., X.W. and W.C.; validation, P.L. and T.Y.; writing—original draft, P.L. and X.X.; writing—review and editing, P.L., X.B. and X.X. All authors have read and agreed to the published version of the manuscript.

**Funding:** This research was funded by the Research and Development of Key Technologies and Complete Sets of Equipment for Recycling Waste Wind Turbine Blades (LHDL-2021-14), the Science and Technology Plan Project of the Department of Housing and Urban-Rural Development of Hubei Province (2023177), and the Natural Science Foundation of Hubei Province (2023AFB245).

**Institutional Review Board Statement:** Not applicable.

**Informed Consent Statement:** Not applicable.

**Data Availability Statement:** No new data were created or analyzed in this study. Data sharing is not applicable to this article.

**Conflicts of Interest:** Peixin Li is employed by the Guodian United Power Technology Co., Ltd. Xiaodan Wang and Weijie Chen are employed by the Guoneng United Power Technology (Baoding) Co., Ltd. Tao Yang is employed by the Wuhan Zhigu Advanced Technology Research Co., Ltd. The remaining authors declare that the research was conducted in the absence of any commercial or financial relationships that could be construed as a potential conflict of interest.

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
