# Peer review of "Recycling of Retired Wind Turbine Blades into Modifiers for Composite-Modified Asphalt Pavements: Performance Evaluation"

_sustainability, doi:10.3390/su16062343_

Round 1

Reviewer 1 Report

Comments and Suggestions for Authors

Line 269.  Figure reference is not correct.

The paper does look at the core aspects of using rWTB in asphalt.  It is presented as a needed recycling, and I agree.  There is value to address the environmental impacts of the use of Saline and Rubber and SBS, which are shown in combination to make the recycling a valuable asphalt additive.  

It would be valuable to show not only virgin asphalt as the comparison, but also asphalt with SBS alone as a comparison material.  Do I get enhanced properties beyond SBS or is the recycling the primary benefit?

Comments on the Quality of English Language

Generally good English use.  Some editing for readablity would improve overall quality.  

Author Response

Point 1: Line 269. Figure reference is not correct.

Response 1: Thanks for your careful reading. The Figure reference is now revised. The following please check the revision:

“Table 7. The attributions of main characteristic peaks checked in Figure 5.”

Point 2: The paper does look at the core aspects of using rWTB in asphalt. It is presented as a needed recycling, and I agree. There is value to address the environmental impacts of the use of Saline and Rubber and SBS, which are shown in combination to make the recycling a valuable asphalt additive.

Response 2: Thanks for your positive comments. The core idea of this paper is to provide a new way to upcycle the WTB into value-added modifiers for enhancing the engineering properties of asphalt pavement.

Point 3: It would be valuable to show not only virgin asphalt as the comparison, but also asphalt with SBS alone as a comparison material. Do I get enhanced properties beyond SBS or is the recycling the primary benefit?

Response 3: Thanks for your comments.

For your first concern, we do include virgin asphalt and SBS modified asphalt for comparisons to check the benefits of WTB modifiers to the asphalt pavement in combination with SBS or alone. In terms of short names, 0R-Si-rWTB/100SBS and 100R-Si-rWTB/0SBS represent the SBS modified asphalt mixture and R-Si-rWTB modified asphalt mixture, respectively.

For your second concern, we can say yes on the enhanced properties of asphalt pavement with the prepared rWTB modifier beyond SBS. Evidences can be found in moisture-induced damage resistance characterized by residual Marshall stability after immersion and residual splitting strength ratio after freeze-thaw treatment, see Figs. 7, 8, 10, and 11. Of course, the appropriate combination of the prepared rWTB modifier and SBS could contribute more to the improved durability of asphalt pavement.

Comments on the Quality of English Language

Generally good English use. Some editing for readability would improve overall quality.

Response: The overall language of the manuscript has been improved a lot from an invited experienced English speaker who studies in our research area.

Reviewer 2 Report

Comments and Suggestions for Authors

Waste of composite materials like glass fibre-reinforced epoxy resins is becoming a serious problem. Particularly, a problem is caused by the rapidly increasing amount of retired wind-turbine blades (rWTR) consisting of such-like epoxy-based composites. Therefore, looking for reusing possibilities of rWTR is urgently necessary. The study of Xu et al. could be a contribution to solve that problem. After crushing/grinding the authors mixed the rWTR powder with styrene-butadiene rubber (SBR), treaded it with a silane-containing reagent and tested its applicability as modifier for asphalt pavements successfully.  

The microstructural properties of the silane and rubber treated rWTR were thoroughly investigated by Scanning Electron Microscopy (SEM) and FTIR spectroscopy. These investigations revealed that the fiber fragments are embedded in the silane-treated rubber. The silane and rubber modified rWTR powder was incorporated into an asphalt mixture. The material obtained this way was investigated by usual tests for asphalt pavements (Immersed Marshall test, Freeze-thaw splitting test, rut depth and dynamic stability, aging stability …). These tests yielded promising results and indicate an applicability of silane/rubber modified rWTR in princible.

That means, the study of Xu et al. yielded interesting findings. In addition, the manuscript is well-written. Therefore, I recommend it for publishing in the journal Sustainability.    

Author Response

Point 1: Waste of composite materials like glass fibre-reinforced epoxy resins is becoming a serious problem. Particularly, a problem is caused by the rapidly increasing amount of retired wind-turbine blades (rWTR) consisting of such-like epoxy-based composites. Therefore, looking for reusing possibilities of rWTR is urgently necessary. The study of Xu et al. could be a contribution to solve that problem. After crushing/grinding the authors mixed the rWTR powder with styrene-butadiene rubber (SBR), treaded it with a silane-containing reagent and tested its applicability as modifier for asphalt pavements successfully.

Response 1: Thanks for your confirmation to this study.

Point 2: The microstructural properties of the silane and rubber treated rWTR were thoroughly investigated by Scanning Electron Microscopy (SEM) and FTIR spectroscopy. These investigations revealed that the fiber fragments are embedded in the silane-treated rubber. The silane and rubber modified rWTR powder was incorporated into an asphalt mixture. The material obtained this way was investigated by usual tests for asphalt pavements (Immersed Marshall test, Freeze-thaw splitting test, rut depth and dynamic stability, aging stability …). These tests yielded promising results and indicate an applicability of silane/rubber modified rWTR in princible.

Response 2: Thanks for your comment.

Point 3: That means, the study of Xu et al. yielded interesting findings. In addition, the manuscript is well-written. Therefore, I recommend it for publishing in the journal Sustainability.

Response 3: Thanks for your comment.

Reviewer 3 Report

Comments and Suggestions for Authors

Innovation: While the idea of recycling rWTB into asphalt pavement modifiers is promising, the manuscript could benefit from a clearer articulation of the novelty of the approach. It would be helpful to discuss how this method differs from existing practices and what unique contributions it brings to the field.

Experimental Design and Analysis: The manuscript would benefit from a more detailed description of the experimental design and analysis methods. Providing more information on how the experiments were conducted and how the results were analyzed would enhance the clarity and robustness of the study.

Significance and Generalizability: The manuscript should further discuss the significance of the findings and their potential impact on the field of road engineering. It would also be valuable to discuss the generalizability of the results to other settings or applications.

Comments on the Quality of English Language

Moderate editing of English language required.

Author Response

Point 1: Innovation: While the idea of recycling rWTB into asphalt pavement modifiers is promising, the manuscript could benefit from a clearer articulation of the novelty of the approach. It would be helpful to discuss how this method differs from existing practices and what unique contributions it brings to the field.

Response 1: Thanks for your comments. The novelty of the design method and the contributions to the field have been added to Introduction section. The following statements have been added:

(1) Third paragraph: To achieve the high-value and high-consumption reuse of rWTB, they are considered as good materials that may enhance the engineering performance of asphalt pavement and address the associated issues in storage and environment. In recent years, there are almost no studies related to its recycling and reuse in asphalt pavement, but considering its material characteristic close to that of FRP, some relevant studies in this aspect can be referenced to for understanding. For example, Lin et al. [18] used GFRP as filler to improve the engineering performance of asphalt mixture, and found that GFRP can improve resistance of asphalt mixture to rut, fatigue, aging, peeling, etc., which, however, is not friendly to the low-temperature performance. Yang et al. [19] also found that FRP, as an asphalt reinforcement material, can not only improve the high-temperature performance of asphalt mixtures, including creep stiffness, rut resistance, and creep-recovery behavior, but also enhance the moisture-induced damage resistance. These similar studies demonstrated that FRP, as a filling material, is proved feasible to be applied in asphalt pavement for its performance enhancement. Therefore, it is believed that rWTB can be potentially reutilized as asphalt modifier to improve the pavement properties.

(2) Fourth paragraph: As well known in asphalt area, SBS is a widely used polymer modifier that can con-tribute to improving the high- and low-temperature properties of asphalt binders and mixtures, but it is very expensive in fact while being used. Considering this, the existing studies have already been published to show whether there are some other cheap materials that can be collectively adopted to replace the incorporation of SBS with no performance compromise of modified mixtures [20,21]. For rWTB, it is normally consisted of glass fiber and epoxy resin, which has been proved effective to improve the overall proper-ties of asphalt pavement and is also considered potential to reduce the use of SBS for satisfactory pavement applications [22]. Therefore, the composite use of rWTB and SBS is very necessary to economically push forward the progress of high-quality asphalt pavement.

(3) Fifth paragraph: For this goal, the current study innovatively proposes a new method on value-added recycling of rWTB into asphalt modifiers through mechanical crushing and grinding in association with further modifications of silane and SBR. Further, this rWTB-based modifier will be considered for single use and combined use with SBS to modify virgin binder for preparing modified asphalt binders and mixtures with different mixing proportions. The microscopic morphology and molecular structure of modifiers will be analyzed, and the properties of rWTB modified mixtures will be checked through water immersion and freeze-thaw tests. Further, their engineering performances, especially moisture-induced damage resistance, will be evaluated after aging. Overall, value-added recycling of rWTB into modifier can contribute to better serving more durable and cost-effective asphalt pavement.

Point 2: Experimental Design and Analysis: The manuscript would benefit from a more detailed description of the experimental design and analysis methods. Providing more information on how the experiments were conducted and how the results were analyzed would enhance the clarity and robustness of the study.

Response 2: Thanks for your comments.

(1) The detailed descriptions of the experimental design to the preparations of modifiers have been further improved. The updated have been revised in the manuscript, which can also be checked from the following:

2.2.1 rWTB modifier

For the first asphalt modifier, it was prepared from a physical method through mechanical crushing and grinding processing. The detailed procedures follow up: (1) first of all, the rWTB was mechanically crushed into flakes with a size of approximately 5-13mm using a crusher; (2) then, these crushed rWTB flakes were ground into powders using a grinder for at least 5min; and (3) the obtained powders were going to the screening device to get the particles sized below 0.3mm for being used as the asphalt modifier. This type of modifier is denoted as rWTB modifier.

2.2.2 Si-rWTB modifier

For the second asphalt modifier, it was obtained from a silane surface treatment method. There are the following steps: (1) anhydrous ethanol and deionized water were mixed at 9:1 by mass ratio to produce mixed solvent; (2) KH550 was added into the mixed solvent with a mass ratio of KH550: ethanol=1:9, and further the mixes were slowly stirred with a glass rod to prepare the hydrolysate; (3) a certain amount of rWTB powder was added to the prepared hydrolysate and mechanically stirred for 30 min at 80℃; and (4) the blends were cured in an oven at 105℃ for 2h, and after grinding, the modified powders were used for bitumen modification. The powders are denoted as Si-rWTB.

2.2.3 R-rWTB modifier

For the third asphalt modifier, it was prepared by considering the adoption of SBR to directly mix with rWTB in a mechanical mixer. The preparation steps are as follows: (1) certain amounts of rWTB and SBR, at a mass ratio of 1:0.3, were manually mixed; (2) the blends were added into a chamber for mechanical mixing at 60℃ for 5min with a shearing speed of 50rpm; and (3) the mixes were then collected for crushing and grinding to get the particles below 0.3mm after screening. These powders, abbreviated as R-rWTB, were used as modifiers for bitumen modification.

2.2.4 R-Si-rWTB modifier

For the fourth asphalt modifier, it was also prepared by considering the adoption of SBR to mix with Si-rWTB in a mechanical mixer. This preparation process is similar to that introduced in Section 2.2.3. The difference is that the base powders to be modified are the Si-rWTB prepared from the process in Section 2.2.2. With regard to this modifier, it is named as R-Si-rWTB.

(2) The detailed descriptions of the analysis methods have been further improved. The updated have been revised in the manuscript, which can also be checked from the following:

2.7.1 Scanning Electron Microscopy (SEM)

To understand the different microstructures of rWTB, Si-rWTB, and R-Si-rWTB modifiers, SEM images were taken for analyzing the treatments. The following steps were conducted: (a) the samples were sprayed with gold at a high vacuum pressure; (b) the samples were placed and fixed into the test chamber; and (c) the images were taken at 500x and 5kx at different sets of test parameters. From microstructural results, the changes in the surface of rWTB particles are needed for observations to analyze whether the silane can work to organically modify the surface of rWTB and the rubber can be better coated onto the surfaces of Si-rWTB.

2.7.2 Fourier Transform Infrared Spectroscopy (FTIR)

This study employed a Nicolet 6700 FTIR spectrometer to check the changes in molecular structure of prepared modifiers, including whether the silane is grafted onto the surface of rWTB and whether the coating of SBR will be removed from the surface of Si-rWTB by mechanical processing. Prior to the test, small modifier particles were mixed with grinded KBr powders to prepare sheet specimens after pressing. During the test, the sheet specimen was first placed to the sample position in the spectrometer chamber and then tested following the conditions which were set at a resolution of 4 cm-1, wavenumber range of 4000-400 cm-1, and scanning time of 16.

2.7.3 Immersed Marshall test

The immersed Marshall test was used in this study to evaluate the effect of different rWTB modifiers on the moisture-induced damage resistance of asphalt mixture. Except for unaged mixtures, the aged mixtures were also tested to indicate if the aging will cause negative impacts on the moisture-induced damage of modified asphalt mixtures. In accordance with JTG E20-2011, the unaged and aged Marshall specimens were immersed in a 60°C water bath for 30min and 48h, and then the Marshall stability values were collected after loading at 50mm/min for determining the residual Marshall stability. The calculation formula is presented in Equation (1):

                           (1)

where, MSr refers to the residual Marshall stability after immersion, %; MSa refers to the Marshall load after immersion at 60°C for 48h, kN; and MSb means the Marshall load after immersion at 60°C for 30min, kN.

2.7.4 Freeze-thaw splitting test

Similar to immersed Marshall test, the freeze-thaw splitting test was also adopted in this study to comparatively evaluate the moisture-induced damage of asphalt mixtures with the incorporation of rWTB modifiers before and after aging. In accordance with JTG E20-2011, the Marshall samples were first prepared by compacting at 50 times for each side, and then approximately 10mL water was added to each sample in closed plastic bags. Before test, these samples were first preconditioned in a refrigerator at -18℃ for 16h and then gone to a water bath at 60℃ for 24h. After completing this, the samples were tested at a loading rate of 50mm/min to record the maximum load for the calculation of the splitting strength. On this basis, the freeze-thaw splitting strength ratio (TSR) can be calculated to characterize the residual resistance of the samples to moisture-induced damage, following the Equation (2):

TSR=×100%                              (2)

where, TSR (%) is the freeze-thaw splitting strength ratio of asphalt mixture; ITS0 is the splitting strength of asphalt mixture without freeze-thaw cycle; and ITS1 is the splitting tensile strength of asphalt mixture after one freeze-thaw cycle.

2.7.5 Wheel tracking test (WTT)

WTT is commonly used to examine the resistance of asphalt mixtures to high-temperature deformation. According to JTG E20-2011, the rut specimens were first prepared with size of 300mm×300mm×50mm through a mechanical rolling method and then placed into the test chamber of rut device for keeping 5h at 60°C. During the test, the specimen was rolled back and forth with a speed of 42 pass/min under a wheel load of 0.7MPa for 60min. After completing, the rut depth data were collected to calculate the dynamic stability (DS) as per Equation (3). With an increase of DS, the resistance of asphalt mixtures to the high-temperature deformation will be enhanced.

DS=                                   (3)

where, DS is the dynamic stability of asphalt mixture, pass/mm; d1 and d2 are rut depth of asphalt mixture at 45min and 60min, mm; and N is the back-and-forth rolling speed of the test wheel, usually 42 pass/min.

Point 3: Significance and Generalizability: The manuscript should further discuss the significance of the findings and their potential impact on the field of road engineering. It would also be valuable to discuss the generalizability of the results to other settings or applications.

Response 3: Thanks for your comments. As suggested, the significance and generalizability of the findings and their potential impact on the field of road engineering are discussed in the section of conclusions. The following contents are added:

“Overall, this study provides a novel recycling and reuse method that can convert rWTB into value-added modifier to enhance the overall engineering performance of asphalt pavement. Further, it has been proved effective to replace partial SBS in binder for durability improvement of mixtures from the evaluations of moisture-induced and aging damages, while contributing to cost savings and environment protections. In fact, the current research results indicated that the prepared modifier shows superior performances to allow asphalt pavement to have more prominent resistances to moisture and freeze-thaw environments for adapting different applications.”

Comments on the Quality of English Language

Moderate editing of English language required.

Response: The overall language of the manuscript has been improved a lot from an invited experienced English speaker who studies in our research area.